# Prenatal Detection of Uniparental Disomies (UPD): Intended and Incidental Finding in the Era of Next Generation Genomics

**DOI:** 10.3390/genes11121454

**Published:** 2020-12-03

**Authors:** Thomas Eggermann

**Affiliations:** Institute of Human Genetics, University Hospital, RWTH Aachen, Pauwelsstr 30, D-52074 Aachen, Germany; teggermann@ukaachen.de; Tel.: +49-241-808-8008; Fax: +49-241-808-2394

**Keywords:** uniparental disomy, prenatal testing, next generation genomics, non-invasive prenatal testing

## Abstract

Prenatal detection of uniparental disomy (UPD) is a methodological challenge, and a positive testing result requires comprehensive considerations on the clinical consequences as well as ethical issues. Whereas prenatal testing for UPD in families which are prone to UPD formation (e.g., in case of chromosomal variants, imprinting disorders) is often embedded in genetic counselling, the incidental identification of UPD is often more difficult to manage. With the increasing application of high-resolution test systems enabling the identification of UPD, an increase in pregnancies with incidental detection of UPD can be expected. This paper will cover the current knowledge on uniparental disomies, their clinical consequences with focus on prenatal testing, genetic aspects and predispositions, genetic counselling, as well as methods (conventional tests and high-throughput assays).

## 1. Introduction

Chromosomal abnormalities significantly contribute to reproductive failure and congenital disorders in humans, and meiotic errors resulting in aneuploidy account for the majority of first trimester pregnancy losses. A result of meiotic or mitotic chromosomal malsegregation is uniparental disomy (UPD) (Figure 1), the exceptional inheritance of the two chromosomes of a pair from the same parent [1], which in contrast to numeric or structural chromosomal aberration, does not affect number and structure of chromosomes. UPD therefore escapes cytogenetic detection, but with the increasing application of high resolution and throughput analysis in diagnostic genetic testing it is detected.

UPD has meanwhile been reported for nearly all human chromosomes. In prenatal diagnosis it can be detected as an incidental finding in the course of genetic tests based on undirected approaches like genome-wide assays. These tests comprise microarray-based molecular karyotyping methods and next generation sequencing (NGS) approaches, and they are increasingly implemented in prenatal genetic testing. In fact, in case of UPD detection, the decision on the further management of the pregnancy requires considerations on methodological aspects, clinical and ethical issues.

## 2. Formation Mechanisms of UPD

Two types of UPD can be discriminated: in uniparental heterodisomy (UPhD), the cell has inherited the two different chromosomes from the same parent, whereas in uniparental isodisomy (UPiD) the same chromosome has been transmitted twice (Figure 1). In fact, complete UPhD or UPiD affecting a whole chromosome can be observed, but frequently stretches of UPhD alternate with UPiD regions on the affected chromosome (mixed UPhD/UPiD).

A major exception is paternal UPD 11 in Beckwith–Wiedemann syndrome (BWS) (Table 1) which is exclusively UPiD. UPD does not always affect a whole chromosome, but also segmental UPDs (also called partial UPD) comprising only segments of a chromosome have been reported (for review: [3]).

UPD can occur after meiotic nondisjunctional errors, but mitotic formation mechanisms have been identified as well (Figure 1):(a)The major formation mechanism is trisomic rescue, with the subsequent loss of one of the supernumerary chromosomes as a prerequisite for the cell to survive. In one third of these reductions, the chromosome from the parent contributing only one chromosome to the trisomic rescue is lost, thus giving rise to a UPD.(b)Gamete complementation describes the fertilization of a disomic gamete by a nullisomic gamete, and is therefore also the result of meiotic errors. The result is complete UPhD.(c)Survival of a monosomic gamete is possible by monosomic rescue via endoduplication, resulting in complete UPiD.(d)Postfertilization UPD formation consists of mitotic malsegregations, either causing a monosomic cell with subsequent endoduplication, or an endoduplication followed by a chromosomal loss. In any case, postzygotic UPD is a complete UPiD.

Depending on the time of meiotic or mitotic nondisjunction, UPDs can occur as a mosaic constitution, even in combination with trisomic cell lines (Figure 1). Accordingly, detection of UPDs can be significantly hampered by the co-occurrence of chromosomal mosaicism, and UPDs might therefore escape detection. 

## 3. Risk Factors Predisposing for UPD Formation

By trisomic rescue, every human chromosome can be affected by UPD, and UPD has meanwhile been reported for nearly all human chromosomes (for review: Liehr T. 2020. Cases with uniparental disomy http://cs-tl.de/DB/CA/UPD/0-Start.html (accessed 11 May 2020)). Accordingly, the frequency of single UPDs correlates with that of the respective trisomies in humans. In fact, trisomy 16 is the most common autosomal trisomy in humans [22] and nearly all trisomy 16 pregnancies originate from maternal meiosis errors [23]. It is therefore not surprising that maternal UPD of chromosome 16 (upd(16)mat) occurs relatively frequently (for review: [24]). 

The link between trisomy and UPD is also reflected by increased maternal age in UPhD in comparison to UPiD (e.g., for upd(7)mat [25]), as increased maternal age is a relevant risk factor for trisomy formation, and so it is for UPD formation. 

Up to 65% of UPDs are detected in individuals with a normal karyotype (46, XX; 46, XY) [26], and the others are associated with chromosomal rearrangements. As these variants predispose to an improper chromosomal segregation in meiosis and mitosis, they can be regarded as another risk factor for UPD formation. In principle, nearly every type of rearrangement can cause a UPD (for review: [27]), but in particular in families transmitting Robertsonian translocations (RobTk) are at risk for UPD formation. RobTks result from whole arm rearrangements of the five acrocentric chromosomes (chromosomes 13, 14, 15, 21, 22) and have an incidence of 1 in 1,000 in the general population (for review: [28]). Due to this frequency and the increased risk of RobTk carriers for trisomy and subsequent UPD formation, in particular, UPDs of the clinically relevant chromosomes 14 and 15 are constantly identified in routine genetic testing. Another group of structural variants prone to UPD formation are small supernumerary isochromosomes, but these are not restricted to specific chromosomes [29].

## 4. Clinical Consequences of UPD 

Up to now, two ways to impact the phenotype have been reported for UPD, i.e., by disturbing imprinted chromosomal regions and by reduction in recessive alleles to homozygosity (Figure 2). As a third way, the possible association with aneuploid cell lines with impact on the phenotype is worthy of mention.

The most prominent clinical outcome of UPDs are imprinting disorders (Table 1). Imprinting disorders are caused by alterations of the balanced monoallelic and parent-of-origin specific expression of imprinted genes [30], and UPDs belong to the spectrum of molecular disturbances in this group of congenital disorders. Though more than 100 human genes are regulated by genomic imprinting, they tend to cluster, and therefore UPD of only some chromosomes and chromosomal regions are associated with imprinting disorder phenotypes. Clinically, imprinting disorders are heterogeneous, but they share several features affecting growth, metabolism and psychomotoric development. In prenatal context, growth disturbances, abdominal wall defects and polyhydramnios might be indicative for disturbed imprinting but are of course unspecific. Though the contribution of UPDs to the molecular spectrum of imprinting disorders vary between the different chromosomes, it should be noted that UPDs were the first molecular alterations which have been described for some of the disorders (e.g., Silver–Russell syndrome, Temple syndrome, Kagami–Ogata syndrome (KOS14), Mulchandani–Bhoj–Conlin syndrome).

As not all human chromosomes harbor imprinted regions, UPDs of single chromosomes can remain without clinical consequence. However, the risk of UPiD in general lies in the reduction in heterozygosity for recessive pathogenic variants to homozygosity, and the first case of molecularly proven UPD was a patient with upd(7)mat, suffering from cystic fibrosis due to homozygosity of p.F508del in the *CFTR* gene [31]. In the meantime, a considerable number of patients with unexpected homozygosity due to UPiD has been reported. Thus, identification of UPiD for a chromosomal segment in association with clinical features might help to identify the molecular cause of the disease based on homozygosity for a recessive pathogenic variant (e.g., [32,33]).

As UPD can be associated with trisomy mosaicism due to its mode of formation; clinical features might also be caused and/or modified by the presence of a trisomic cell line. In principle, this possibility applies for all UPDs, but depending on the formation mechanism and the gene content of the affected region, trisomy mosaicism does not play a relevant role. However, in some imprinting disorders mosaicism for the UPD is correlated with the severity of the disease (e.g., upd(11)pat in BWS). In the classical imprinting disorders, trisomy mosaicism can be neglected to have a clinical impact, but there are at least two chromosomal constitutions for which the pathogenetic influence of either the trisomy mosaicism or the UPD is discussed: for upd(16)mat, the clinical heterogeneity has been attributed to mosaic trisomy 16 cell lines [17], but the recent identification of a case with an isolated methylation defect [18] indicate that at least some features might be linked to an imprinting defect. In case of upd(6)mat, the impact of trisomy 6 cell lines has been suggested [5] whereas further evidence for an imprinting effect has not yet been published.

## 5. Techniques to Detect UPDs

Techniques to detect UPDs are generally based on genetic polymorphisms, and the segregation analysis of polymorphic alleles in the patient and their parents. Thus, the final proof of UPD can only be obtained by the comparative analysis of the genotype of the patient with those of their parents. In classical UPD testing, short tandem repeat (STR, microsatellite) analysis is the most often used tool as it can be conducted easily at a low price. STRs are DNA stretches containing repeat units of between two and seven nucleotides in length that are tandemly repeated. Therefore, this class of markers has a high information value. With the exception of single STRs associated with trinucleotide repeat disorders, their length is transmitted steadily from generation to generation, and can therefore be used to trace the transmittance of alleles and chromosomal regions (Figure 3a). UPD can also be identified by other marker systems like single nucleotide polymorphisms (SNPs), but these analyses require the analyses of a larger number of markers as SNPs commonly only consist of two alleles and the information content of a single SNP is much lower that of an STR.

However, this problem is circumvented by assays which allow the analysis of a huge number of markers in parallel. Examples are SNP arrays (i.e., used for molecular karyotyping) (Figure 3b) and NGS-based assays. Both allow the detection of stretches of loss of heterozygosity (LOH, i.e., stretches of homozygosity), which might be caused by UPiD. In addition, comparative analyses of the parental genotypes and application of specific bioinformatics tools allow the detection of UPhD as well [34,35]. In fact, this comparison enables the final proof of UPD in the majority of cases, but in families with a high degree of consanguinity, LOH might not be distinguishable from UPD.

In families suspicious for UPD affecting imprinted loci, this problem can be circumvented by the use of methylation-specific (MS) tests in some situations. An example is parallel testing of the two oppositely imprinted loci in 11p15.5 in Beckwith–Wiedemann syndrome; gain of methylation of the telomeric imprinting control region (IC1) and loss of methylation of the centromeric region (IC2) in the same patient confirms the molecular diagnosis of a paternal upd(11p) (Figure 3c).

In summary, the commonly used tests for UPD detection can be discriminated in those used for targeted detection of UPD (STR typing, MS tests), and those providing information on UPD as an incidental finding (SNP array, NGS).

## 6. UPDs and Prenatal Testing

In a prenatal context, UPD is detected either in the course of a directed testing workup for UPD, or incidentally (Figure 2).

The first group commonly comprises pregnancies in families with a known chromosomal aberration predisposing for UPD formation, e.g., RobTks. Though UPD formation can occur in the course of nondisjunction of any chromosomal aberration, it is very rare and the risk figures are heterogeneous. In fact, the major group at risk for formation of a clinically relevant UPD are RobTk carriers affecting chromosomes 14 and 15. For these structural variants, risk figures have been determined showing that carriers of inherited as well as de-novo RobTks can exhibit UPD [36]. In RobTK families, the risk for UPD formation in a fetus with non-homologous RobTk has been estimated as 0.6–0.8% [36]. 

In rare cases, UPD testing might be indicated in families with imprinting disorders due to chromosomal rearrangements affecting the imprinted regions on the aforementioned chromosomes (Table 1). In these cases, the type and consequence of alteration (causing disturbance of the imprinted region by breakage, deletion or duplication) and the sex of the parent transmitting the alteration has to be considered. In particular, in case of the imprinted chromosomal regions known to be associated with parent-of-origin dependent phenotypes (PWS-AS, BWS-SRS, TS14-KOS14) different clinical outcomes can be observed if the affected chromosome is inherited either from the mother from the father (e.g., [37]).

Incidental identification of a UPD can occur in the course of different prenatal settings, and the implementation of high-resolution assays have boosted this observation. The group of incidental diagnoses of UPDs can be subdivided in two modes of ascertainment, (a) identification via prenatal detection of (numerical) chromosomal aberrations, and (b) by unexpected homozygosity (and stretches of LOH).

As described before (Figure 1), the major formation mechanism of UPD is trisomy rescue, and therefore UPD detection after trisomy rescue is a well-known phenomenon in classical invasive prenatal cytogenetic analyses. In the past, the trisomy was typically identified in chorionic villous sampling (CVS), but due to the lethality of many trisomies, trisomic rescue is the only chance of the fetus to survive, and therefore the trisomy is not detectable in fetal tissues (e.g., amniotic cells) (confined placental mosaicism (CPM)). However, in the course of this rescue, UPD might occur. In case of chromosomes which do not harbor imprinted genes or recessive pathogenic alleles, these UPDs remain undetected, but they are identified in case clinically relevant sequences are affected. Therefore, it has been suggested to conduct UPD testing in pregnancies with chromosomal aberrations affecting chromosomes 6, 7, 11, 14, 15 and 20 [38], though it should be carefully considered in respect to the decision on the further progress of the pregnancy [39].

## 7. Detection of UPD in the Era of Prenatal High-Resolution Testing

Prenatal UPD detection has become a topic again with the implementation of non-invasive prenatal testing (NIPT) on the basis of NGS-based analyses of fetal cell free DNA in maternal serum. In fact, the first cases of UPD have already been proven after detection of trisomy/CPM by NIPT ([40,41], own unpublished data), and due to the rapid increasing application of NIPT in pregnancy management, the detection of a growing number of UPDs can be expected.

UPD can also be identified incidentally by high-resolution assays seeking genomic causes of fetal malformations and diseases [32,42]. As mentioned before, LOH stretches indicate UPiD [42], and this genetic constitution can therefore help to identify homozygosity for recessive pathogenetic variants, or disturbed inheritance of imprinted regions (e.g., KOS14 [43]). On the other hand, incidental UPD detection by prenatal NGS analysis might indicate a (mosaic) chromosomal disturbance as the cause of clinical features.

Whereas UPiD is detectable by LOH stretches obtained by high-resolution assays in a single patient, UPhd would escape detection. However, UPhD is in principle also detectable by SNP microarray or NGS analyses, but UPhD regions are commonly identified by the comparison of the patients’ genotypes with those from the parents (trio-based analysis in case of NGS).

The application of high-resolution assays in prenatal management significantly improves the detection rate for genomic alterations causing aberrant phenotypes (for review: [44,45]). On the other hand, the ascertainment of an increasing amount of genetic data is accompanied by a growing number of incidental findings, comprising both obviously pathogenic variants as well as variants of unknown clinical significance. In the end, these data might become difficult to interpret and to advice. 

These considerations also refer to UPDs: the incidental identification of a UPD affecting a region harboring clinically relevant imprinted genes requires the comprehensive and state-of-the-art counselling of the family. The situation is further complicated in case a chromosomal region of uncertain imprinting status and clinical relevance is affected. An example is the prenatal identification of upd(16)mat after NIPT for which the clinical consequences are currently not predictable [17]. However, even in case of UPD of chromosomal regions which are not associated with clinical phenotypes due to disturbed imprinting, there is the risk of homozygosity for a recessive pathogenic allele which cannot be estimated. Thus, the identification of a UPD without obvious clinical relevance might contribute to uncertainty of the advice-seeking couple without providing seminal knowledge.

Despite these challenges in (prenatal) diagnosis, the detection of UPD by high-resolution assays also provides further knowledge on UPD and its functional relevance. It allows the identification of segmental UPDs and thereby narrows down clinically relevant imprinted regions, and uniparental disomic regions might help to identify new genes and recessive variants [46]. In contrast to single locus tests (e.g., STRs), the high-throughput and high-resolution assays allow the genome-wide identification of (segmental) UPDs. 

## 8. Conclusions and Outlook

The implementation of next generation genomic assays with a considerable high-resolution (e.g., microarrays, next and third generation sequencing, genome imaging) in the genetic diagnostic setting has already been conducted, and numerous studies and publications have proven the power, appropriateness and reliability of the different methods to identify SNVs and copy number variants (CNVs) as well as epigenetic alterations. In addition, they allow the identification of UPD, thereby the functional and clinical relevance of the whole chromosome as well as segmental UPDs can be elucidated. In parallel to the validation of next generation genomic assays in postnatal genetic diagnostics, their application in prenatal management has been pursued. In fact, prenatal chromosome microarray analysis [44] and NIPT are meanwhile widely used screening tools, and prenatal NGS to identify monogenetic causes of fetal malformations is in establishment [47]. With the increased application of these approaches as first-line tests, the number of incidental findings will further increase, and these findings comprise UPDs as well. However, as shown in this review, the incidental identification of UPD is often difficult to manage, and should therefore be embedded in a comprehensive multidisciplinary management approach, in particular, it should include a high-quality ultrasound and clinical assistance. It must be embedded in genetic counselling, and ethical issues have to be addressed so that the patients can finally come to a self-determined decision.

Generally, pre-test counselling is also necessary to inform on the benefits, limitations and diagnostic scope of genetic tests.

## Figures and Tables

**Figure 1 genes-11-01454-f001:**
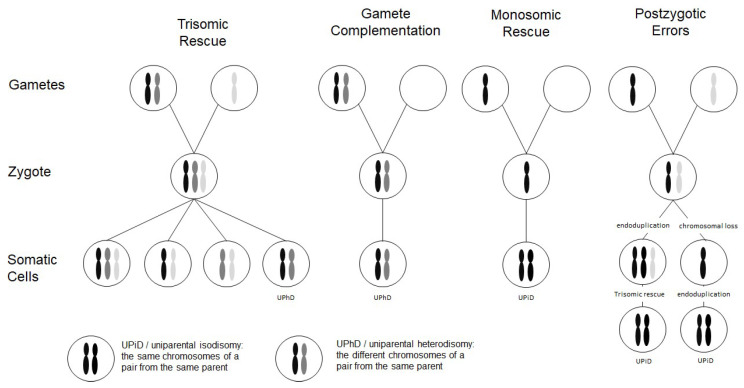
Scheme depicting the four major formation mechanisms of whole chromosome uniparental disomies (UPDs). As it can be delineated from these modes of formation, mosaicism of a UPD cell line as well as an aneuploidy cell line is possible in case of trisomic rescue and postzygotic errors (for further details see [2]). It should be noted that in case of monosomic rescue and postzygotic nondisjunction, uniparental idisomy of the whole chromosome can be observed, whereas in the other modes of formation, uniparental heterodisomy can be interrupted by stretches of uniparental disomy. Formation of segmental UPD as well as UPDs caused by structural chromosomal variants (like Robertsonian translocation) are not shown (for that issue see [3]).

**Figure 2 genes-11-01454-f002:**
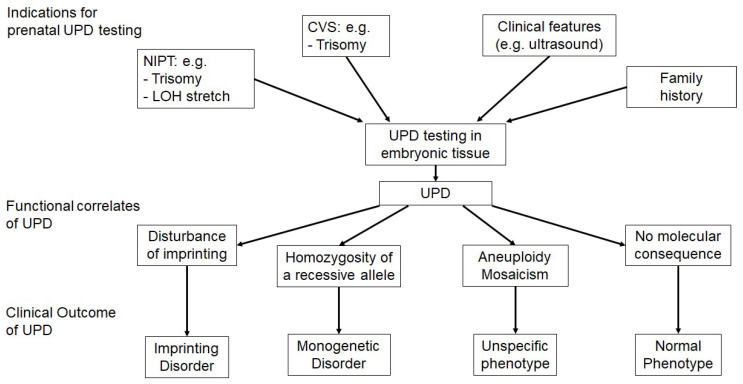
Overview on the indications for uniparental disomy (UPD) testing, the functional correlates and clinical outcomes of UPD, (NIPT—non-invasive prenatal testing, LOH—loss of heterozygosity, CVS—chorionic villous sampling).

**Figure 3 genes-11-01454-f003:**
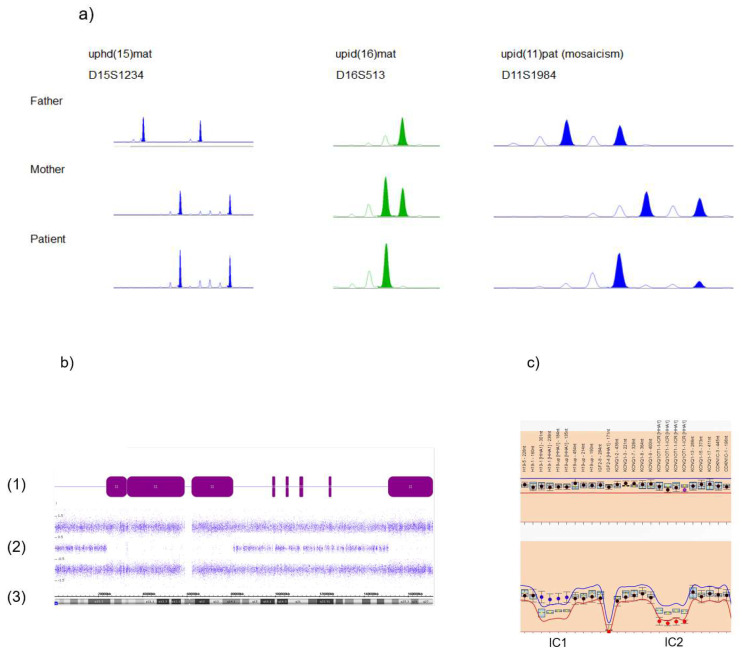
Standard molecular methods to identify uniparental disomy (UPD). (**a**) Short tandem repeat (STR) typing illustrating uniparental heterodisomy (UPhD) (upd(15)mat: D15S1234), uniparental isodisomy (UPiD) (upd(16)mat: D16S513) and mosaic upd(11)pat in a BWS patient (marker D11S1984; mosaicism for uniparental isodisomy of 11p15 can be delineated by the relatively weak maternally inherited allele, whereas the inherited paternal allele shows a much stronger intensity). (**b**) Detection of segmental UPiD of chromosome 6 by single nucleotide polymorphism (SNP) array analysis (Cytscan SNP array, Affymetrix, Santa Clara, CA, USA): (1) stretches of loss of heterozygosity (LOH) (purple bars, corresponding to UPiD), interrupted by non-LOH stretches probably representing UPhD; (2) B allele frequencies; (3) ideogram of chromosome 6 and physical basepair position. (**c**) Identification of the upd(11)pat by methylation-specific multiplex ligation-dependent probe amplification (MS MLPA), (upper panel: copy number analysis, lower panel: methylation analysis). The patient exhibits a normal copy number in 11p15.5, thereby excluding deletions or duplication as the cause for the aberrant methylation affecting the imprinting centers 1 and 2 (IC1, IC2). Methylation analysis shows a hypermethylation of the paternally methylated IC1 and hypomethylation of the maternally methylated IC2, therefore a upd(11)pat can be delineated (based on MS MLPA using the ME030-C3 assay and data analysis by the Coffalyser.Net software, MRC Holland, Amsterdam, NL, USA).

**Table 1 genes-11-01454-t001:** Uniparental disomies (UPDs) for which an association with congenital disorders is known or discussed.

UPD	Imprinting Disorder OMIM	Prevalence of the Imprinting Disorder in General	Fraction of UPD among the Molecular Disturbances	Prenatal Features	Major Postnatal Clinical Features	Mosaicism Reported	Reference
Upd(6q24)pat	Transient neonatal diabetes mellitus (TNDM) 601,410	1/300,000	41%	IUGR, macroglossia, abdominal wall defects	transient diabetes mellitus, hyperglycemia without ketoacidosis, macroglossia, abdominal wall defects	No	[4]
Upd(6)mat *	In discussion	unknown	unknown	IUGR, oligohydramnios	Hernia, PNGR, heterogeneous findings	Trisomy 6	[5]
Upd(7)pat	In discussion	Unknown	unknown	NR	Overgrowth (1/5 cases)	No	[6]
Upd(7)mat	Silver–Russell syndrome (SRS) 618,905	1/75,000–1/100,000	5–10%	IUGR	PNGR, relative macrocephaly at birth, body asymmetry, prominent forehead, feeding difficulties, learning difficulties	No	[7]
Upd(11p15)pat	Beckwith–Wiedemann syndrome (BWS) 130,650	1/15,000	20%	Macroglossia, exomphalos, (lateralized) overgrowth, maternal preeclampsia	Macroglossia, exomphalos, lateralized overgrowth, Wilms tumor or nephroblastomatosis, hyperinsulinism, adrenal cortex cytomegaly, placental mesenchymal dysplasia, pancreatic adenomatosis	Yes	[8,9]
Upd(11p15)mat	Silver–Russell syndrome (SRS) 180,860	1/75,000–1/100,000	Single case	IUGR	PNGR, relative macrocephaly at birth, body asymmetry, prominent forehead, feeding difficulties, learning difficulties	Yes	[7,10]
Upd(14q32)pat	Kagami–Ogata syndrome (KOS14) 608,149	unknown	65%	IUGR, polyhydramnion, abdominal wall defects, bell-shaped thorax, coat-hanger ribs	abdominal wall defects, bell-shaped thorax, coat-hanger ribs, developmental delay	No	[11]
Upd(14q32)mat	Temple syndrome (TS14) 616,222	unknown	29%	IUGR	PNGR, neonatal hypotonia, feeding difficulties in infancy, truncal obesity, scoliosis, precocious puberty, small feet and hands	No	[12,13,14]
Upd(15q11.2)pat	Angelman syndrome (AS) 105,830	1/20,000 −1/12,000	3–7%		Severe intellectual disability, microcephaly, no speech, unmotivated laughing, ataxia, seizures, scoliosis	No	[15]
Upd(15q11.2)mat	Prader–Willi syndrome (PWS) 176,270	1/25,000 −1/10,000	20–25%	IUGR	PNGR, Intellectual disability, neonatal hypotonia, hypogenitalism, hypopigmentation, obesity, hyperphagia	No	[16]
Upd(16)mat *	In discussion	Unknown	Unknown	IUGR	Cardial, vascular signs, skeletal signs, hernia; SRS features	Trisomy 16	[17,18]
Upd(20q13)pat	Pseudohypo-parathyroidism 1B (PHP1B) 603,233	Unknown	2–3%		Resistance to PTH and other hormones, Albright hereditary osteodystrophy, subcutaneous ossifications, feeding behavior anomalies, abnormal growth patterns	No	[19]
Upd(20)mat	Mulchandani–Bhoj–Conlin syndrome (MBCS) 617,352	Unknown	100%	IUGR	PNGR, feeding difficulties	NR	[20,21]

* In these UPDs it is discussed whether the phenotype is caused by the UPD or mosaicism for the aneuploidy; IUGR—intrauterine growth retardation, PNGR—postnatal growth retardation; NR—not reported.

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
