# Peer review of "Prenatal Detection of Uniparental Disomies (UPD): Intended and Incidental Finding in the Era of Next Generation Genomics"

_genes, 2020, doi:10.3390/genes11121454_

Round 1

Reviewer 1 Report

The author reviews the origin of unipartental disomies and their detection in prenatal diagnosis. The article is well organized and used references are correct. I just have some minor comments:

- Between some text words there is an extra space, eg page 5 line 85 (trisomy formation), page 8 line 189 (these structural).

- Page 8 line 210, "However" should be "however"

- Figure 1, please explain RobTK

- Figure 3b) indicate the number of each subfigure

Author Response

The author reviews the origin of unipartental disomies and their detection in prenatal diagnosis. The article is well organized and used references are correct. I just have some minor comments:

- Between some text words there is an extra space, eg page 5 line 85 (trisomy formation), page 8 line 189 (these structural).

ANSWER: Done

- Page 8 line 210, "However" should be "however"

ANSWER: Done

- Figure 1, please explain RobTK

ANSWER: Done

- Figure 3b) indicate the number of each subfigure

ANSWER: Done

Reviewer 2 Report

Manuscript: Prenatal detection of uniparental disomies (UPD): Intended and incidental finding in the era of next generation genomics.

Minor concerns:

  1. Figures: Every figure should be independently readable. So every aberration should be spelled out the full names when firstly appear. Like "LOH" in figure 3, though the full name has been spelled out in the text.
  2. In the text, every aberration should be spelled out the full names when firstly appear. Like in line 113 SRS, TS14, KOS14, MBCS. Those full names only are spelled out in table 1.

Author Response

Minor concerns:

  1. Figures: Every figure should be independently readable. So every aberration should be spelled out the full names when firstly appear. Like "LOH" in figure 3, though the full name has been spelled out in the text.

ANSWER: Done

  1. In the text, every aberration should be spelled out the full names when firstly appear. Like in line 113 SRS, TS14, KOS14, MBCS. Those full names only are spelled out in table 1.

ANSWER: I´m sorry for the inconsistency, and I´ve changed this. 

Reviewer 3 Report

I would congratulate with the author who performed a clear and accurate review about the current knowledge on prenatal detection of UPD.The manuscript is well written and nicely put together several useful considerations for managing prenatal genetic counseling. Just some typing/formatting errors across the text (eg. I would use the lowercase after colon and I would try to better format the table1).

Author Response

I would congratulate with the author who performed a clear and accurate review about the current knowledge on prenatal detection of UPD.The manuscript is well written and nicely put together several useful considerations for managing prenatal genetic counseling. Just some typing/formatting errors across the text (eg. I would use the lowercase after colon and I would try to better format the table1).

ANSWER: I thank the reviewer for this stimulating comment, and I´ve check the draft for the comments.